# Estimating the effects of legalizing recreational cannabis on newly incident cannabis use

**Barrett Wallace Montgomery**[1]*, **Meaghan H. Roberts**[2], **Claire E. Margerison**[1], **James C. Anthony**[1]

**1** Department of Epidemiology and Biostatistics, College of Human Medicine, Michigan State University, East Lansing, MI, United States of America, **2** Department of Economics, College of Social Science, Michigan State University, East Lansing, MI, United States of America

* montg270@msu.edu

**Data Availability Statement:** The data relevant to this study is available from Github at https://github.com/Predict-This/Recreational-Cannabis-Leagalization.

## Abstract

Liberalized state-level recreational cannabis policies in the United States (US) fostered important policy evaluations with a focus on epidemiological parameters such as proportions [e.g., active cannabis use prevalence; cannabis use disorder (CUD) prevalence]. This cannabis policy evaluation project adds novel evidence on a neglected parameter–namely, estimated occurrence of newly incident cannabis use for underage (<21 years) versus older adults. The project's study populations were specified to yield nationally representative estimates for all 51 major US jurisdictions, with probability sample totals of 819,543 non-institutionalized US civilian residents between 2008 and 2019. Standardized items to measure cannabis onsets are from audio computer-assisted self-interviews. Policy effect estimates are from event study difference-in-difference (DiD) models that allow for causal inference when policy implementation is staggered. The evidence indicates no policy-associated changes in the occurrence of newly incident cannabis onsets for underage persons, but an increased occurrence of newly onset cannabis use among older adults (i.e., >21 years). We offer a tentative conclusion of public health importance: Legalized cannabis retail sales might be followed by the increased occurrence of cannabis onsets for older adults, but not for underage persons who cannot buy cannabis products in a retail outlet. Cannabis policy research does not yet qualify as a mature science. We argue that modeling newly incident cannabis use might be more informative than the modeling of prevalences when evaluating policy effects and provide evidence of the advantages of the event study model over regression methods that seek to adjust for confounding factors.

## Introduction

In drug dependence epidemiology, the estimated prevalences of active drug use are population health statistics that hide important patterns of (a) incidence (occurrence of first onsets) and (b) duration (e.g., duration and frequency of use after it starts). Lapouse [1], building upon prior work [2], argued that incidence estimates tell us about causes. In contrast, prevalence

**Funding:** There was no research support from the cannabis or other non-federal or non-university sources. BWM, MHR, CEM, and JCA wish to acknowledge support from the Michigan State University Vice President for Graduate Studies and Research (university funds) as well as federal research grant support from the National Institutes of Health (5R25DA051249). BWM and MHR also wish to acknowledge the Michigan State University Graduate School for funding from the Graduate Enrichment Fellowship and the University Distinguished Fellowship, respectively. The funders had no role in study design, data collection and analysis, decision to publish, or preparation of the manuscript. There was no additional external funding received for this study.

**Competing interests:** The authors have declared that no competing interests exist.

estimates tell us about caseloads and health services burdens. In a more recent review of the substance use epidemiology literature, Wu and colleagues echo these sentiments and note an abundance of research on prevalence, but a lack of literature on incidence [3].

Cheng and colleagues exploited this incidence-prevalence differentiation to show that a large sub-population of young adults in the United States (US) deliberately delayed their first drink until after the legal minimum drinking age [4, 5]. Prevalence hid this pattern. Members of our research group hypothesized that age-specific cannabis use incidence would show a similar pattern developing in jurisdictions that legalized cannabis: Once the legal minimum age for recreational cannabis use was set at 21 in some states, many young adults will wait until cannabis use is legal for them to try it [6].

These initial observations motivated this research to estimate whether legalizing recreational cannabis might affect the occurrence of newly incident cannabis use (i.e., incidence). Cannabis use incidence in the US has traditionally peaked between ages 15 and 17 with steady declines as each cohort gets older [6]. Since all states that legalized recreational cannabis set 21 as the legal minimum age to purchase recreational cannabis, we analyze incidence before and after the age 21 milestone is reached. We sought to understand how legalizing recreational cannabis may be affecting incidence for these two age strata and how the estimates can inform the US population experiences after cannabis policy liberalization.

We can see no prior research on cannabis use incidence post recreational cannabis legalization (RCL). The published literature to this point has evaluated prevalence of recent use, prevalence of cannabis use disorder (CUD), and frequency of use. Concerning associations between cannabis liberalization and cannabis use prevalence among youth, most published evidence indicates that prevalence did not change after legalization, and perhaps may have dropped in some sub-[7, 8, 9, 10, 11, 12]. Yet, a minority of studies provide firm evidence of appreciable cannabis use prevalence increases among adolescents [13, 14, 15, 16]. As for CUD prevalence, the published evidence indicates that the 12-17-year-old participants in the National Surveys on Drug Use and Health (NSDUH) in states with legalized recreational cannabis might have been more likely to be CUD cases, but causal attribution to cannabis policy change remains uncertain [17]. As for the frequency of cannabis use among adolescents, the published estimates show no changes post RCL [17, 18, 19, 20].

Among adults of legal age to purchase cannabis in these states, the evidence looks quite different. Apart from a few early findings [15, 21, 22], the published estimates consistently show that the prevalence of cannabis use among adults may increase after legalization [9, 10, 17]. Increased odds of CUD were found among NSDUH respondents 26 and older [17] and poly use of cannabis with other drugs, including alcohol, was also found to have increased in adults over the age of 26 [23]. Nevertheless, other studies find no evidence of change and deem the evidence to be inconclusive [10, 20]. One study described an increase in frequent use in the 26 and older age group, but in no other sub-groups [17]. Another study found no increase in frequent or daily use in any sub-group [10].

To add novelty to cannabis policy evaluation research, we turned to the event study framework, an extension of the classic differences-in-differences (DiD) model. The DiD model is popular when the research goal is to estimate causal policy effects in the context of policy interventions in which the exposure and control groups are likely to differ on many dimensions. Its popularity might be traced to its constraints on unobserved confounding variables with the framework of relatively loose assumptions that the contrasted observed trends are parallel [24]. The event study model extension defines periods before and after legalization as intervention leads and lags. These lead and lag indicators allow for dynamic modeling of estimated changes in cannabis use incidence before and after the intervention.

We sought to estimate the causal effect of US state cannabis policy liberalization on the occurrence of newly incident cannabis use with respect to the legal minimum age. We produced age-stratified estimates for underage population members who were prohibited from purchasing cannabis, and for adults who were allowed to purchase retail cannabis, in several time periods relative to the dates of legalization.

## Methods

### Study population and sample

For this study, the population was specified to include non-institutionalized US civilian residents, sampled and assessed for successive NSDUH survey waves, 2008 through 2019. These NSDUH cross-sectional surveys were conducted with multistage area probability sampling to draw state-level representative samples and to over-sample 12-to-17-year-olds. The total sample size for surveys conducted in this period includes 819,543 respondents. The average weighted screening participation level for the sample was 82% with an average interview participation level of 71% [25]. As this research used publicly available and anonymized data, the research was determined as not human subjects research by the Michigan State University Institutional Review Board on 8/26/2021 (MSU Study ID: STUDY00006620).

Standardized audio computer-assisted self-interview modules assessed each newly incident user's month and year of first cannabis use, from which incidence estimates were derived from the NSDUH Restricted Data Access portal (R-DAS). R-DAS estimates are analysis-weighted with Taylor series derived variances and 95% confidence intervals (CI). The R-DAS portal also allows for state-specific analysis of data but can only be downloaded in pairs of years and not individual years (e.g., 2018–2019 vs. 2018, 2019). Thus, we produce estimates from six year-pairs in these analyses, not from 12 individual years.

We categorized states into different analysis groups according to each state's year of legalization through 2018. Because the 2018–2019 year-pair is the most recent available data in R-DAS at the time of analysis, states that legalized cannabis in 2019 or later were categorized into the control group in which retail cannabis remained illegal. Washington and Colorado were included in the 2012 group. Oregon, Alaska, and Washington D.C. were in the 2014 group. California, Maine, Massachusetts, and Nevada were included in the 2016 group. Vermont and Michigan were included in the 2018 group. All other states were categorized into the control group for this analysis.

### Primary outcome

Our primary estimate is the occurrence of newly incident cannabis use, calculated as $\psi = X_r/N_r$, where $X_r$ is the number of individuals starting to use cannabis within the 1–12 month interval before assessment (NSDUH variable RECMJ_B until 2013, RECMJ2 starting in 2014) and $N_r$ is all persons who had not started using cannabis before that interval (NSDUH variable ELIGMJ_B until 2013, ELIGMJ2 starting in 2014). Prevalences are estimated as $p_1 = X_r/N$, where $N$ is the total projected population size, and the estimated proportion of the population at risk ($p_2 = N_r/N$), with the corresponding standard errors. Proportions of newly incident cannabis use are estimated from $p_1$ and $p_2$ as:

$$\psi = \frac{p_1}{p_2} = \frac{X_r/N}{N_r/N}$$

## Study design and statistical analysis

Recent explorations and analyses by econometricians revealed that estimating an average treatment effect is a bit of an over-simplification, especially when policy adoption is staggered [26, 27, 28, 29]. With a policy intervention described as a 'treatment', the average treatment effect on the treated (ATT) is a weighted average of all the possible two-period estimators. This estimate can be problematic if it averages out important treatment effect heterogeneity that can take place over time. If treatment effects vary over time, then the ATT estimate is biased [26].

We found some evidence that drug policy intervention effects might change over time due to these lagged policy effects, thus we believe the event study model is better suited to this context [30, 31]. Our study design contrasts estimates of cannabis incidence in the RCL states relative to non-RCL states before and after the legalization of cannabis at the state level. The DiD event study modelling yields estimates in each period relative to the year prior to legalization while controlling for fixed differences across states and national trends over time.

Our models can be expressed as:

$$Y_{st} = RCL_s \times \sum_{\substack{y = -5 \\ y \neq -1}}^{4} \beta_y I(t - t_s^* = y) + \beta_t + \beta_s + \epsilon_{st}$$

As described earlier, our datasets are constructed at the state category ($s$) by year ($t$) level. In our primary analyses, $Y_{st}$ denotes the cannabis incidence estimate for each state grouping in each year-pair. In the equation, $\beta_s$ denotes state fixed effects and $\beta_t$ denotes the fixed effects of time in calendar years. As a result, general time trends in cannabis incidence for each group of states are accommodated.

The variable $RCL_s$ is set equal to one if the observation is from a state that legalized cannabis with measurements before after the date of legalization and is set equal to zero otherwise. Time-event dummy variables $I(t - t_s^* = y)$ indicate the legality of cannabis in each state group by the first year of the R-DAS year-pair relative to the year of legalization ($t_s^*$) and are set equal to zero for all observations from states that did not legalize recreational cannabis during the study period. These variables are referred to in this analysis as 'leads' (indicators of time-event before legalization) and 'lags' (indicators of time-event after legalization). The omitted category is $y = -1$, the year-pair before legalization. Therefore, each $\beta_y$ estimate quantifies the difference in newly incident cannabis use occurrences in the RCL states relative to states with no policy change during year $y$ compared to differences in the year-pair that immediately preceded legalization. When only one or two categories of states would be included at an interval because of the variation in legalization timing across states ($\leq 6$ years before legalization and $\geq 4$ years after legalization), some lead and lag indicators are combined to balance the extremes and prevent modelling the outcome for only small subsets of the data. This is commonly referred to as balancing the leads and lags of the model [27].

If occurrences of newly incident cannabis use trend similarly in all groups before legalization, we would expect that the estimated coefficients for the lead indicators will be small and indifferent from the null value in a test of the parallel trends assumption built into our model. When estimated coefficients for the lag indicators are positive departures from the null, this provides supporting evidence to reject the null hypothesis (e.g., an increase in the occurrence of newly incident cannabis use in RCL states).

In addition to the event study estimates of change at each time interval, we also present a simple 2x2 DiD estimate of the ATT as a summary of the estimated effect on those aged 21 and older across all post-legalization years through 2019 and an average treatment effect with the

same method for the 12-to-20-year-olds. This estimate is derived from the same equation with the event study dummy variables replaced with a single indicator for post-policy change states.

**Dates of legalization vs. dates of implementation.**   We note that the mean number of days between the date of legalization and actual retail sales in the states in our sample (except for Washington D.C. where sales have never been legal) is approximately 500 days [32]. We set the T0 interval for this study to be a close approximation of this interval of elapsed time between policy enactment and actual implementation (i.e., start of retail sales).

**Alternative specifications and robustness checks.**   To ensure the robustness of our analyses, we examined two alternate specifications. The first alternate specification uses the same method to estimate the effect of RCL on cannabis prevalence. The estimate for prevalence has been studied extensively in the literature and we compare our results to prior estimates as a check of face validity for our model. The second robustness check uses a time placebo as a check of robustness. In this model, a random year within the data was selected as the year that states legalized cannabis. The model is then run with the same specifications. If any of this model's coefficients are large enough to reject the null hypothesis, the evidence suggests a potentially spurious relationship.

All beta coefficients from the models are multiplied by 100 for interpretation as percent changes in the one-year cumulative incidence proportions. All analyses were performed in SAS version 9.04 with NSDUH analysis weights and Taylor series variances.

## Results

### Descriptive statistics

In aggregate, the population sample under study included 819,543 respondents from the NSDUH surveys conducted between the years 2008 and 2019. The unweighted sample distributions indicate 48% female, 60% White, 13% Black, 18% Hispanic, 2% Native American, 4% Asian, and 4% of more than one race or another race or ethnicity (Table 1). Within the sample, 11% used cannabis recently (past month). Table 1 provides the total unweighted sample characteristics with the NSDUH Public Data Analysis System (P-DAS) used to derive these values.

S1–S5 Figs show cannabis use incidence estimates for those aged 21 and older over time in different combinations of the state legal categories. Upon visual inspection, the parallel lines assumption and assumption of no anticipation look to have been met in every group by group comparison. For the sake of context and comparison, the average proportion of newly incident cannabis use between 2008 and 2019 in states that never legalized cannabis is 6.2% for 12-to-20-year-olds and 0.5% for those aged 21 and older. The average proportion of newly incident cannabis use in the two years prior to legalization for states that did legalize cannabis is 7.8% for 12-to-20-year-olds and 0.9% for those aged 21 and older.

### Event study findings

Figs 1 and 2 show the primary findings for individuals aged 21 and older (Fig 1) and those between the ages of 12 and 20 (Fig 2). For those who were legally able to purchase cannabis (21 and older), the legalization of cannabis is estimated to have had no effect on newly incident cannabis use in the years of legalization. However, between two and four years after legalization, RCLs are estimated to have increased incidence by 0.6% [95% Confidence Interval (CI) = 0.1, 1.0]. The corresponding estimate for the interval four to seven years after passage of the RCL is 1.3% [0.8, 1.8] (Fig 1). For the 12-to-20-year-olds, the estimated cannabis incidence does not vary appreciably in any period (Fig 2).

**Table 1. Characteristics of the U.S. population under study from the U.S. National Surveys on Drug Use and Health.**

| Gender | % | n |
|---|---|---|
| Female | 47.8% | 322,636 |
| Male | 52.2% | 351,885 |
| Race | | |
| White | 59.9% | 404,314 |
| Black | 12.8% | 86,272 |
| Native American | 1.5% | 10,095 |
| Native Hawaiian / Other Pacific Islander | 0.5% | 3,380 |
| Asian | 4.1% | 27,907 |
| More than one race | 3.6% | 24,301 |
| Hispanic | 17.5% | 118,252 |
| Age | | |
| 12–17 Years Old | 28.1% | 189,789 |
| 18–25 Years Old | 29.0% | 195,650 |
| 26–34 Years Old | 12.7% | 86,000 |
| 35 or Older | 30.1% | 203,082 |
| Past month cannabis use prevalence | | |
| Did not use in the past month | 88.7% | 597,984 |
| 4Used within the past month | 11.3% | 76,537 |
| Unweighted Sample Total | 100.0% | 674,521 |

## DiD findings

When including the total time post-legalization, the simple ATT estimate derived from the 2x2 DiD indicates no substantial differences in cannabis incidence before and after the laws were passed (p = 0.12). However, since we expected no effect before cannabis sales became effective, we estimated a separate ATT for two years of legalization and later in the 21+ age group as 0.7% (p = 0.003, [0.3, 1.1]). The estimated average treatment effects for those aged 12 to 20 years indicated no differences after the legalization date (p = 0.27) or the effective date (p = .53).

## Alternative specifications and robustness checks

In our first alternate specification, we estimate that the effect of cannabis legalization increased the prevalence of cannabis use in the past month among those aged 21 and older by 3.2% between two and four years after legalization (p = 0.0005, [1.6, 4.7]). The corresponding estimate for the interval four to seven years after legalization is 4.3% (p = 0.0002, [2.3, 6.2]) (S6 Fig). In the 12-to-20-year-old age group, no appreciable variation in estimated cannabis use prevalence is seen across these study intervals (P = 0.39 and 0.33, respectively) (S7 Fig).

In the time placebo analysis based upon a randomized legalization date, the date of placebo legalization was set to the year 2011 for all the states that legalized cannabis through 2018. S8 Fig shows an estimated coefficient that does increase slightly over time, yet the estimated effect of this 'placebo' policy change is null. Note especially that for the adolescents (<21 years), the coefficients are distributed more or less at random in relation to the zero value, with no appreciable differences or patterns (S9 Fig).

## Discussion

These results show consistent evidence of an increase in the occurrence of newly incident cannabis use for adults aged 21 years and older after the removal of prohibitions against cannabis

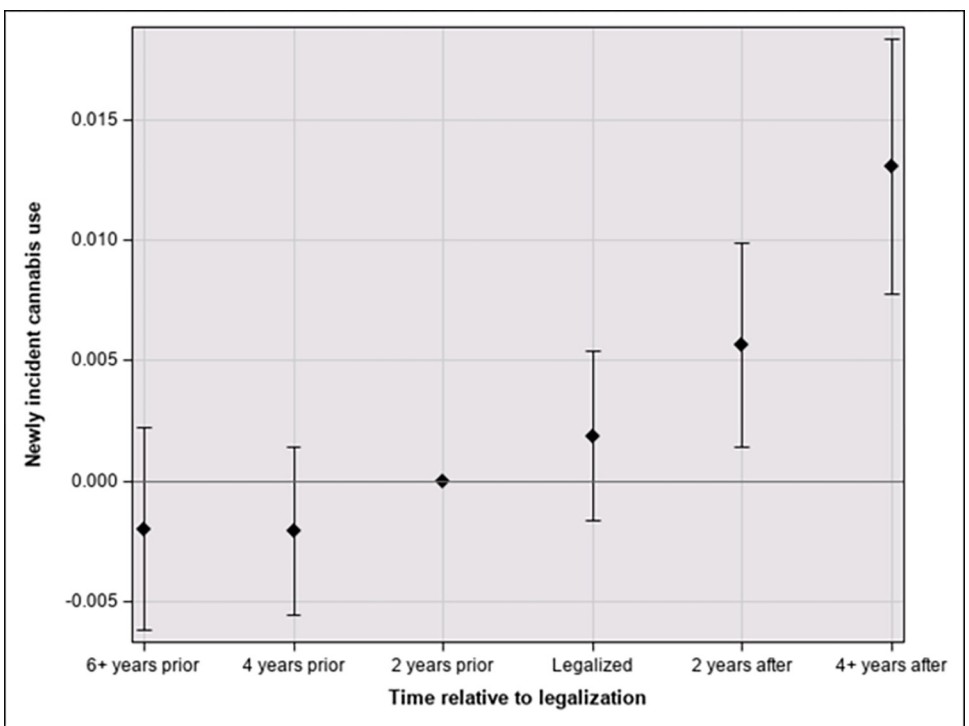

**Fig 1. Effect of time since cannabis legalization on cannabis incidence in the 21 and older age group with 95% confidence intervals.**

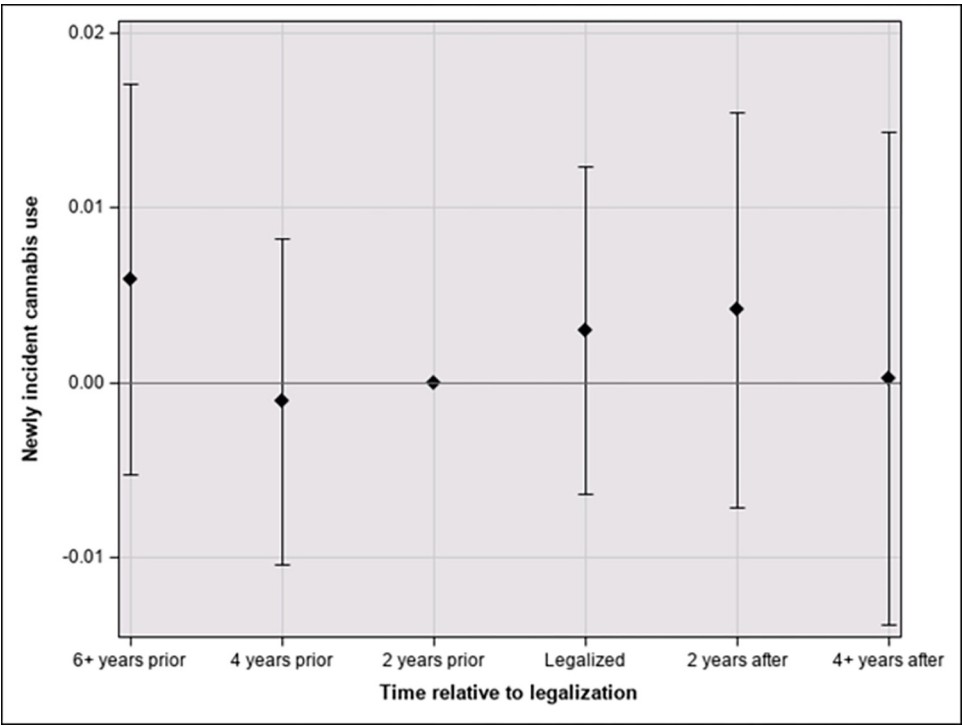

**Fig 2. Effect of time since legalization on incidence in 12-to-20-age-group with 95% confidence intervals.**

retail sales. For those aged 12-20-years-old, the study estimates support the hypothesis that RCLs did not affect the occurrence of newly incident cannabis use for underage persons. In the simple 2x2 DiD models, we estimate an average increase in cannabis use incidence of 0.7 percentage points after recreational cannabis began being legally sold through the year 2019, nearly double the difference between these state groups pre-legalization.

The innovations of this policy analysis relative to prior efforts can be seen in several areas. First, we focus on occurrence of newly incident cannabis use, separating out the population of sustained cannabis users. Prior studies on the associations between RCLs and cannabis use epidemiology focused on past-month cannabis use prevalence [7, 8, 9, 10, 13, 16, 19, 21, 22], the prevalence of daily or frequent users [19, 11, 19], and prevalence of CUD [10, 17]. As such, the importance of understanding changes in cannabis use incidence in response to legalizing recreational cannabis cannot be overstated. Prevalence of use and dependence syndromes and frequency of use are of great public health importance, yet they tell us nothing about whether new users are entering into the population of cannabis users. This study provides an important initial thread of evidence about how liberalized cannabis policies might affect the number of cannabis users who otherwise might never have tried the drug.

Second, our research approach allows for the possibility raised by Cheng and colleagues [4, 5] with respect to alcohol and by Montgomery, Vsevolozhskaya, & Anthony with respect to cannabis [6]. That is, there might exist a large pool of law-abiding individuals who would never have used cannabis if retail sales had not been allowed, but who try cannabis once it becomes legal for them to do so.

Third, this is the first study of which we are aware that has examined the heterogeneity of treatment effects in the years post RCL. The event study design allows for the estimation of effects by years relative to the passage of the recreational cannabis legislation and the effective dates of implementation. This has resulted in three important pieces of evidence: 1) Estimated effects of cannabis legalization on incidence of use seems to increase over time (albeit with possible diminishing returns); 2) Estimated effect sizes vary across age strata defined by the legal minimum retail sales age; and 3) Estimated effect size might be zero for the population to whom cannabis remains illegal. This last piece of evidence might provide some reassurance to policy makers who worry about increased incidence among adolescent populations of the jurisdictions that permit cannabis purchases by adults.

Fourth, the use of a quasi-experimental DiD design provides some allowance for a causal interpretation of estimated intervention effects. With some noteworthy exceptions [11, 13], the evidence published on cannabis policy effects has relied mostly on controlling for observed variables between the populations. Considerable differences exist between populations in states with and without legalized recreational cannabis. It seems reasonable to ask whether controlling for pre-contemplated and measured variables is sufficient to produce valid estimates. The DiD framework constrains unobserved variables within a limited framework of model-based assumptions. Our research included evaluation of some of these often-untested assumptions (e.g., no anticipation; parallel trends).

Lastly, due to our focus on cannabis incidence, this study's estimates cannot be compared directly with findings of prior cannabis policy evaluations. Nonetheless, a limited comparison is possible and can be seen in the results from our application of the DiD approach to estimates of the prevalence of cannabis use. As in the estimates published by Cerdá and colleagues [13] and by Coley and colleagues [11], our DiD approach disclosed no appreciable policy influence on cannabis prevalence estimates for people under the age of 21. Our estimates of prevalence are similar to the estimates seen in Cerdá et al., Martins et al., and Reed's more recent findings ([17, 10, 9]. We also note that our findings may help the field to understand seemingly conflicting earlier null findings in this age [15, 21, 22]. Synthesizing the above findings, we suggest

that the increases in the use of cannabis in the adult age group may have only began increasing after a few years when recreational cannabis shops began sales.

## Limitations and strengths

Before describing some directions for future cannabis policy research, we must describe several limitations of our empirical study. First, it's difficult to conceptualize cannabis policy evaluation studies that do not rely upon self-reports from general population samples. In other domains, we might look to retail sales records, but before cannabis policy shifts to permit retail sales there are no pre-policy measurements. We also might look to employer records on drug-testing of employees, but these databases are selective and non-representative of the larger population experience, without coverage of the important age strata we have studied. It seems unlikely that cannabis policy evaluation research will overcome the self-report as a limitation for the time being. As an extension of this concern about self-report, we must acknowledge the possibility of differential response biases. Might population members be more likely to disclose cannabis use when they can use cannabis without concern about legal consequences? This question has yet to be answered. The assessments were conducted using confidential standardized audio computer-assisted self-interview modules which have been shown to reduce biases of this type.

Some other limitations of this work include the sensitivity of the findings to different definitions of the study period and an inability to control for sub-state level recreational cannabis legality. The limitation regarding the definition of the study period is important, specifically to our estimate of the ATT. When including the two-year period immediately after legalization (before sales began) in the treatment period, we detected no differences. However, using a study design that allows for dynamic treatment effects and having estimates that are robust to alternate specifications allow us to show where and when the difference in trends occur. This supports the argument that the effect of cannabis legalization is driven by the opening of outlets where recreational cannabis is sold.

Another limitation of this work at the state level is that many counties and municipalities within states that have legalized recreational cannabis have chosen to ban the sale or cultivation of cannabis within that sub-state area. For example, in Washington State, 15% of counties and 55% of municipalities have prohibited the sale of cannabis [33] while in California, 69% of counties and 70% of cities prohibit the sale [34]. Like the null finding between the date of legalization and effective dates of cannabis sales, we expect that estimates of the effects of legalizing recreational cannabis at the state level are diminished by incorporating incidence for many individuals who reside in areas where recreational cannabis is effectively in this pre-implementation state. This sub-group heterogeneity is averaged out in our state-level estimates. While a sub-state analysis is beyond the scope of this study, future research should seek to replicate this analysis at the municipality or county level.

The strengths of this work are the robustness of the estimates, the novelty of the design in this space, and the interpretations that it allows for. Our estimates of the effects of recreational cannabis liberalization on cannabis use incidence by age group were robust to both the check of face validity using the same method to estimate past-month prevalence and the alternate specification using a time-placebo analysis. The use of the DiD event study design moves this field forward by allowing for a dynamic estimate of the causal effect of RCL on the outcome of choice.

As we have demonstrated, it is not reasonable to assume that the effect of cannabis legalization is homogenous over time, especially not if the period includes the time before cannabis sales began. Therefore, future research on the effects of RCL should allow for time-specific effect heterogeneity. Although this is only one study, from which conclusions should not be

drawn, this design allows for a visualization of the policy lag effect, about which much has been written [30, 31]. We see that the effect is not linear and is perhaps rather sigmoidal in shape with the increases in incidence and prevalence beginning to plateau, although more data is needed to confirm the trend.

## Conclusions

This study contributes novel estimates of how liberalized cannabis policies within US jurisdictions might have influenced occurrence of newly incident cannabis use in the underage (<21 years) and in the adult populations, now allowed to purchase cannabis products in retail outlets. Cannabis policy liberalization continues to be a contentious issue in the national political landscape with different risks and benefits described for all of the potential paths forward. Policy-makers and the voters who elect these policy-makers cannot make the best judgments in the absence of evidence, unless their decisions are to be based on potentially erroneous prejudices or beliefs. The evidence from this study is not perfect, but the estimates provide an evidence base that can be judged in relation to an important question–namely, should we worry about underage cannabis use when adults are allowed to buy cannabis products in retail shops? And might the occurrence of adult-onset newly incident cannabis use increase if this policy change is made? The answer to the first question at this point seems to be that there has been no policy influence on cannabis incidence in the underage adolescent population after adults have been allowed to buy cannabis in retail shops. The answer to the second question at this point indicates a tangible uptick in the occurrence of newly incident cannabis use among adults who otherwise might never have tried cannabis. We are hopeful that voters, policy-makers, and public health officials can use this evidence as they forecast what might change if cannabis policies are liberalized to permit adult purchases from retail cannabis shops in their jurisdictions.

## Supporting information

**S1 Fig. Cannabis incidence in 21 and older age group, first wave legalizing states vs untreated states.**
(PDF)

**S2 Fig. Cannabis incidence in 21 and older age group, second wave legalizing states vs untreated states.**
(PDF)

**S3 Fig. Cannabis incidence in 21 and older age group, third wave legalizing states vs untreated states.**
(PDF)

**S4 Fig. Cannabis incidence in 21 and older age group, first wave legalizing states vs third wave legalizing states.**
(PDF)

**S5 Fig. Cannabis incidence in 21 and older age group, second wave legalizing states vs third wave legalizing states.**
(PDF)

**S6 Fig. Effect of time since cannabis legalization on past month cannabis prevalence in the 21 and older age group.**
(PDF)

**S7 Fig. Effect of time since legalization on past-month cannabis prevalence in the 12-to-20-age-group.**
(PDF)

**S8 Fig. Placebo effect of time since cannabis legalization on cannabis incidence in the 21 and older age group.**
(PDF)

**S9 Fig. Placebo effect of time since cannabis legalization on cannabis incidence in the 12-to-20-age-group.**
(PDF)

## Author Contributions

**Conceptualization:** Barrett Wallace Montgomery, James C. Anthony.

**Formal analysis:** Barrett Wallace Montgomery, Meaghan H. Roberts.

**Investigation:** Barrett Wallace Montgomery.

**Methodology:** Barrett Wallace Montgomery, Meaghan H. Roberts, Claire E. Margerison.

**Project administration:** Claire E. Margerison.

**Supervision:** Claire E. Margerison, James C. Anthony.

**Validation:** Meaghan H. Roberts.

**Visualization:** Barrett Wallace Montgomery.

**Writing – original draft:** Barrett Wallace Montgomery.

**Writing – review & editing:** Barrett Wallace Montgomery, Meaghan H. Roberts, Claire E. Margerison, James C. Anthony.

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
