## [Decision Letter · Decision Letter 0]

27 Apr 2022

PONE-D-22-06272Estimating the Effects of Legalizing Recreational Cannabis on Newly Incident Cannabis UsePLOS ONE

Dear Dr. Montgomery,

Thank you for submitting your manuscript to PLOS ONE. After careful consideration, we feel that it has merit but does not fully meet PLOS ONE’s publication criteria as it currently stands. Therefore, we invite you to submit a revised version of the manuscript that addresses the points raised during the review process.

We look forward to receiving your revised manuscript.

Kind regards,

Giuseppe Carrà, PhD

Academic Editor

PLOS ONE

Journal Requirements:

(There was no research support from the cannabis or other non-federal or non-university sources. 

BWM, MHR, CEM, and JCA wish to acknowledge support from the Michigan State University Vice President for Graduate Studies and Research (university funds) as well as federal research grant support from the National Institutes of Health (5R25DA051249). BWM and MHR also wish to acknowledge the Michigan State University Graduate School for funding from the Graduate Enrichment Fellowship and the University Distinguished Fellowship, respectively.

The funders had no role in study design, data collection and analysis, decision to publish, or preparation of the manuscript.)

4. We noted in your submission details that a portion of your manuscript may have been presented or published elsewhere.  (PLOSM , presubmission inquiry, 

PMEDICINE-D-22-00520) Please clarify whether this publication was peer-reviewed and formally published. If this work was previously peer-reviewed and published, in the cover letter please provide the reason that this work does not constitute dual publication and should be included in the current manuscript.

6. Please include your tables as part of your main manuscript and remove the individual files. Please note that supplementary tables (should remain/ be uploaded) as separate "supporting information" files"

Reviewers' comments:

Reviewer's Responses to Questions

**Comments to the Author**

1. Is the manuscript technically sound, and do the data support the conclusions?

Reviewer #1: Yes

Reviewer #2: Partly

2. Has the statistical analysis been performed appropriately and rigorously? 

Reviewer #1: I Don't Know

Reviewer #2: I Don't Know

3. Have the authors made all data underlying the findings in their manuscript fully available?

Reviewer #1: Yes

Reviewer #2: Yes

4. Is the manuscript presented in an intelligible fashion and written in standard English?

Reviewer #1: Yes

Reviewer #2: Yes

5. Review Comments to the Author

Reviewer #1: Studies evaluating the impacts on cannabis use of more liberal state-level cannabis policies in the United States have focused on outcomes such as the prevalence of cannabis use and cannabis use disorder (CUD) in national surveys of the general population and youth. The authors argue that these studies have neglected the impact of these policy changes on the incidence of cannabis use, especially among young people whose cannabis use is a major community concern. They report estimates of incidence of cannabis use among those under the legal purchase age (<21 years) compared this with the incidence among older adults.

They note that under prohibition the incidence of cannabis use in the US peaked between ages 15 and 17 with a steady decline as each cohort aged. Because all US states that have legalized recreational cannabis have set 21 as the legal purchase age, the authors wondered whether incidence may vary before and after the population reaches the age of 21 years. milestone is reached. They assessed whether legalizing recreational cannabis affected the incidence of cannabis use in those under and over the age of 21 years.

The authors used data on cannabis use collected annually from nationally representative US samples in all 51 major jurisdictions between 2008 and 2019. Their total sample comprised a probability sample of 819,543 non-institutionalized US residents. They answered standardized questions on cannabis use in computer-assisted interviews, including questions related to the onset of cannabis use in the past year that were used to estimate incidence. They modelled difference in-difference in incidence to study the effects of policy changes in different US states where policy implementation differed over time while controlling for fixed differences between jurisdictions.

Their analysis did not find any policy-associated increase in newly incident cannabis use among persons under the age of legal cannabis purchase age (21 years) in states that had legalised adult use. By contrast, they found an increase in the incidence of new cannabis use among older adults over the age of 21 years and this occurred with a lag after retail sales commenced.

The authors tentatively conclude that legalizing cannabis retail sales in US states increased cannabis use among older adults, but not among those under the legal retail purchase age. They argue that this approach to modeling the effects of legal policy changes on newly incident cannabis use might be more informative for policy makers and the public than modeling the impacts of policy changes on the prevalence of use or cannabis use disorders.

The authors have provided a novel approach to evaluating the impact of cannabis legalisation on uptake among persons who are under and over the minimum legal purchase age in states that have legalised adult cannabis use. Their analysis also presents evidence that setting the minimum legal purchase age at 21 years has encouraged a substantial law abiding section of young people to delay initiating cannabis use until after they have reached the legal purchase age.

Reviewer #2: The approach is novel here and of interest. Given the limitations of state-level RCL, discussed below, it is not entirely clear to me what new information the study contributes and how it will move the field. I think that the method, and focus on initiation, are unique. Yet, if the conclusion is that there is more initiation among adults over age 21 as cannabis is ‘legalized,’ and not among those under 21, consistent with some but not all prior findings (see below) I would question whether this finding will hold up as time goes on and/or whether it is valid now. Specifically, the authors hint at this, but it does not seem to be fully discussed, but could it not be that no increase is being seen among those for whom use remains illegal simply due to that very fact and their mistrust of reporting illegal behavior? If not for all, this is very likely to lead to an undercount among particular subgroups who may be particularly distrustful of government or any organization which could be related to law-enforcement.

The field is increasingly moving toward the understanding that State-level RCL has tremendous limitations in terms of evaluating impact on cannabis use because of the heterogeneity of exposure within a state. This applies not only at a local level (e.g., density and proximity to outlets) but even state-wide because within a given state, for instance, California is a large state by population and geography. The majority of municipalities prohibit the sale of recreational cannabis. Yet, those that do are often densely populated with retail outlets. The authors note this in the limitations section and suggest a county-level approach next. As a side note, cannabis policy is not made at the “county” level in many states, perhaps excepting WA and CO, but at sub-county (town, village, other type of municipality level). Additionally, one RCL state is not comparable to another RCL state in many respects, aside from the general endogeneity issue. The way RCL is implemented, rather than the RCL itself, appears to affect use. Features of RCL that differ within and between states are numerous and impactful, and have been shown to have more impact than RCL itself in policy studies.

So, I find that the results of a state-level analysis, especially up through and including only 2019—which is a very short timeframe post-RCL adoption, is very limited in terms of impact and real-world implications. Further, RCL seems fairly unstoppable at this point but so ways to create policy that is optimal in terms of maximum benefit (not only to big cannabis companies) and minimal harm could have more impact at this point.

The conclusion that there is no increase in cannabis incidence among youth seem also to conflict with prior findings of several studies, so I would be curious how the authors reconcile these findings with those.

On a more technical note, the way that incident cannabis use was defined using the NSDUH variables was not clear to me. I read it several times. Maybe including the specific variables used, just making it more explicit, would be helpful for readers.

Would it potentially be possible to model both incidence and past 30-day use outcomes using this alternative method (which I found compelling) to examine whether results are consistent or whether they diverge.

I am not clear on what the authors mean by allowing for ‘heterogeneity’ that should be considered with RCL effects in this context? Among various groups or only over time? I believe it is clear there is a policy lag not only because that is natural, but because there is often a lengthy delay between “passage/adoption” of a RCL which then leads to more immediate “decriminalization” prior to dispensary openings. For instance, NJ has had a lag of several years, I think, or something along those lines.

In sum, I appreciate the novel approach and the potential import of incidence. But I am not sure that this moves the field farther along at this point given the issues mentioned above.

I’ve included several references that seem relevant with the hope that they may be helpful. The conclusions cited regarding whether there have been increases in use among 12-17 year olds (or those under age 21) overall and/or by RCL status seem not as consistent as suggested, potentially.

Goodwin RD, Pacek LR, Copeland J, Moeller SJ, Dierker L, Weinberger AH, Gbedemah M, Zvolensky MJ, Wall MM, Hasin DS. Trends in daily cannabis use among cigarette smokers in the United States, 2002-2014. American Journal of Public Health, 2018; 108: 137-142.

Weinberger AH, Zhu J, Lee J, Anastasiou E, Copeland J, Goodwin RD. Cannabis use among youth in the United States, 2004-2016: Faster rate of increase among youth with depression. Drug and Alcohol Dependence, 2020.

Weinberger AH, Wyka K, Kim JH, Mangold M, Smart R, Schanzer E, Wu M, Goodwin RD. A difference-in-difference approach to examining the impact of Cannabis legalization on disparities in the use of cigarettes and cannabis in the United States, 2004-2017. Addiction, 2022.

Kim JH, Weinberger AH, Zhu J, Barrington-Trimis J, Wyka K, Goodwin RD. Impact of state-level cannabis legalization on poly use of alcohol and cannabis in the United States, 2004-2017. Drug and Alcohol Dependence, 2021.

6. PLOS authors have the option to publish the peer review history of their article (what does this mean?). If published, this will include your full peer review and any attached files.

Reviewer #1: **Yes: **wayne hall

Reviewer #2: No

---

## [Author Response · Author response to Decision Letter 0]

23 May 2022

Reviewer and Comment # Comment Response

Journal Requirement #1 When submitting your revision, we need you to address these additional requirements.

Thank you for providing these guiding documents on formatting. The manuscript and associated files have been formatted according to the journal’s requirements. 

References have been changed to adhere to the Vancouver style

Journal Requirement #2 Please provide additional details regarding participant consent. In the ethics statement in the Methods and online submission information, please ensure that you have specified what type you obtained (for instance, written or verbal, and if verbal, how it was documented and witnessed). If your study included minors, state whether you obtained consent from parents or guardians. If the need for consent was waived by the ethics committee, please include this information. Thank you for this suggestion, I added the following sentence to the first paragraph on the methods study population: “As this research used publicly available and anonymized data, the research was determined as not human subjects research by the Michigan State University Institutional Review Board on 8/26/2021 (MSU Study ID: STUDY00006620)”

Journal Requirement #3 Thank you for stating in your Funding Statement: 

(There was no research support from the cannabis or other non-federal or non-university sources. 

BWM, MHR, CEM, and JCA wish to acknowledge support from the Michigan State University Vice President for Graduate Studies and Research (university funds) as well as federal research grant support from the National Institutes of Health (5R25DA051249). BWM and MHR also wish to acknowledge the Michigan State University Graduate School for funding from the Graduate Enrichment Fellowship and the University Distinguished Fellowship, respectively.

The funders had no role in study design, data collection and analysis, decision to publish, or preparation of the manuscript.)

Please include your amended Funding Statement within your cover letter. We will change the online submission form on your behalf. Added the sentence “There was no additional external funding received for this study.” in the updated Funding Statement and included it in the cover letter

Journal Requirement #4 We noted in your submission details that a portion of your manuscript may have been presented or published elsewhere. (PLOSM , presubmission inquiry, 

PMEDICINE-D-22-00520) Please clarify whether this publication was peer-reviewed and formally published. If this work was previously peer-reviewed and published, in the cover letter please provide the reason that this work does not constitute dual publication and should be included in the current manuscript. The work was not peer-reviewed. It was published online in the medrxiv pre-print server: doi: https://doi.org/10.1101/2022.01.26.22269900

Journal Requirement #5 Please include your full ethics statement in the ‘Methods’ section of your manuscript file. In your statement, please include the full name of the IRB or ethics committee who approved or waived your study, as well as whether or not you obtained informed written or verbal consent. If consent was waived for your study, please include this information in your statement as well. The following sentence was added to the first paragraph of the methods section: “As this research used publicly available and anonymized data, the research was determined as not human subjects research by the Michigan State University Institutional Review Board on 8/26/2021 (MSU Study ID: STUDY00006620)”

Journal Requirement #6 Please include your tables as part of your main manuscript and remove the individual files. Please note that supplementary tables (should remain/ be uploaded) as separate "supporting information" files" Thank you for the suggestion, table 1 is included in the main manuscript on page 12.

Journal Requirement #7 Please include captions for your Supporting Information files at the end of your manuscript, and update any in-text citations to match accordingly. Please see our Supporting Information guidelines for more information: http://journals.plos.org/plosone/s/supporting-information. Added the required page for Supporting information with corrected supplemental figure titles. Updated in-text citations to match.

Reviewer 1 #1 None We thank reviewer one for a succinct and accurate summary of our work.

Reviewer 2 #1 could it not be that no increase is being seen among those for whom use remains illegal simply due to that very fact and their mistrust of reporting illegal behavior? We thank the reviewer for bringing up this important limitation. We discussed this issue as an extension of the limitations of self-report in our section on the limitations of this research. To reiterate: this is absolutely a possibility that we acknowledge, but it has yet to be shown empirically. However, past research has shown that cannabis is one of the most reliably and valid questions in these surveys and the standardized audio computer-assisted self-interview modules were designed to reduce biases of this type. We did add sub-section headings in our discussion section to more clearly emphasize where the limitations of this study are discussed. 

Reviewer 2 #2 As a side note, cannabis policy is not made at the “county” level in many states, perhaps excepting WA and CO, but at sub-county (town, village, other type of municipality level) Thank you for this note. Our reading of the policy literature on this topic shows that states (as you reference) differ in terms of the level of government which is allowed jurisdiction over the issue. In another reading, we do see that you are correct that we do seem to use the term county as a stand in for both counties and municipalities. We have changed some of the language in the manuscript to be more inclusive of the other levels of government that regulate this policy.

Reviewer 2 #3 The conclusion that there is no increase in cannabis incidence among youth seem also to conflict with prior findings of several studies, so I would be curious how the authors reconcile these findings with those. To be clear, no publication to date has studied the incidence of new cannabis use. Therefore, there cannot be a conflict in findings regarding incidence. We did show no increase in past 30 day prevalence among those aged 12-20. However, in our reading of the literature, studies that showed an increase in cannabis use prevalence among youth were in the minority and tended to be earlier papers, some of which used the date of legal change instead of the date of the effective change (i.e., Cerda et al 2017 and Paschall, García-Ramírez, & Grube, 2021). The prevalence of cannabis use disorder among youth increased in both studies that used that outcome, but that is a very different outcome.

Reviewer 2 #4 On a more technical note, the way that incident cannabis use was defined using the NSDUH variables was not clear to me. I read it several times. Maybe including the specific variables used, just making it more explicit, would be helpful for readers. Thank you for this helpful suggestion, we have added the NSDUH variables required for this analysis. 

Reviewer 2 #5 Would it potentially be possible to model both incidence and past 30-day use outcomes using this alternative method (which I found compelling) to examine whether results are consistent or whether they diverge. We thank the reviewer for this suggestion. We did in fact model both incidence as well as past 30 day use. The past 30 day use outcomes was used as a robustness check in the way the reviewer suggested. However, the results are somewhat tucked-away in that section of the results and included in the supplemental figures S6 and S7.

Reviewer 2 #6 I am not clear on what the authors mean by allowing for ‘heterogeneity’ that should be considered with RCL effects in this context? Among various groups or only over time? I believe it is clear there is a policy lag not only because that is natural, but because there is often a lengthy delay between “passage/adoption” of a RCL which then leads to more immediate “decriminalization” prior to dispensary openings. We thank the reviewer for this helpful clarification. In most cases in this paper, we were referring to the heterogeneity over time as suggested by the reviewer (i.e., policy lag). We modified some of the language in the paper to make this reference clearer. 

Reviewer 2 #7 I’ve included several references that seem relevant with the hope that they may be helpful. The conclusions cited regarding whether there have been increases in use among 12-17 year olds (or those under age 21) overall and/or by RCL status seem not as consistent as suggested, potentially. We that the reviewer for these suggested articles and have added the two that were relevant to recreational cannabis policy associations and effects to our summary of the literature.

---

## [Editor Report · Decision Letter 1]

7 Jul 2022

Estimating the Effects of Legalizing Recreational Cannabis on Newly Incident Cannabis Use

PONE-D-22-06272R1

Dear Dr. Montgomery,

We’re pleased to inform you that your manuscript has been judged scientifically suitable for publication and will be formally accepted for publication once it meets all outstanding technical requirements.

Kind regards,

Giuseppe Carrà, PhD

Academic Editor

PLOS ONE

---

## [Editor Report · Acceptance letter]

12 Jul 2022

PONE-D-22-06272R1 

Estimating the Effects of Legalizing Recreational Cannabis on Newly Incident Cannabis Use 

Dear Dr. Montgomery:

I'm pleased to inform you that your manuscript has been deemed suitable for publication in PLOS ONE. Congratulations! Your manuscript is now with our production department. 

Kind regards, 

on behalf of

Dr. Giuseppe Carrà 

Academic Editor

PLOS ONE